# The impact of port green competitiveness on the hinterland economy: A case study of China

Yajing Li[1], Ming Yin[1]*, Jiawei Ge[1,2]

1 College of Transport & Communications, Shanghai Maritime University, Shanghai, China, 2 School of Naval Architecture Ocean & Civil Engineering, Shanghai Jiaotong University, Shanghai, China

* 205660@shmtu.edu.cn

**Data Availability Statement:** The data underlying the results presented in the study are available from China Stock Market Accounting Research Database[https://data.csmar.com/] and National Bureau of Statistics [https://data.stats.gov.cn/].

## Abstract

In the global trade landscape, ports are not only critical nodes for the transportation of goods but also increasingly recognized as key factors driving sustainable development through their green competitiveness. This paper takes a green competitiveness perspective to delve into the interactive relationship between port development and the hinterland economy, aiming to provide a theoretical foundation and policy recommendations for the green development of ports. An evaluation index system for green competitiveness of ports was constructed, and a combination of the Slacks-Based Measure (SBM) and the Entropy Weighted-Technique for Order Preference by Similarity to Ideal Solution (EW-TOPSIS) method was employed to quantitatively measure and assess the 25 major coastal ports and their hinterland economic systems in China. The study reveals that the overall green competitiveness of these major coastal ports is on the rise, yet notable disparities exist among individual ports. Regionally, the green competitiveness of the Yangtze River Delta port group stands out, while that of the Bohai Rim, southeastern, Pearl River Delta, and southwestern coastal port groups is relatively low. Empirical analysis conducted in this paper demonstrates the positive impact of port green competitiveness on the hinterland economic system, particularly its significant role in enhancing the total and quality aspects of the hinterland economy. Furthermore, it was found that the green competitiveness of ports exerts a more pronounced promotional effect on the hinterland economies of the Yangtze River Delta and Pearl River Delta port groups compared to that on the Bohai Rim, southeastern, and southwestern coastal port groups. The results of this paper also hold important guiding value for optimizing the allocation of port resources, promoting port technological innovation, and improving port environmental management, contributing to the harmonious coexistence and long-term development of ports and their hinterlands.

## 1 Introduction

The competitiveness of a port signifies its comparative advantage over other ports in terms of capturing the transportation market, adding value to logistics, and maintaining sustainable

**Funding:** This research is funded by National Social Science Foundation of China (Grant Number: 20BJY177). The funders had no role in study design, data collection and analysis, decision to publish, or preparation of the manuscript.

development, by relying on its own infrastructure, technology, and other high-quality services to attract more elements of investment [1, 2]. As a crucial node in international trade, enhancing its competitiveness is of significant importance to the development of the national economy. The enhancement of port competitiveness not only attracts more cargo transportation but also promotes the economic development of the region where the port is located.

With the expansion of port scale, the original development mode of the port, while meeting the needs of regional economic development, has led to many problems such as extensive resource utilization and serious environmental pollution, which in turn limit the further development of the port. To solve these problems, ports in various countries have begun to explore new development paths, including the implementation of green port policies and the integration of environmental protection concepts. As a pioneer, the Long Beach Port in the United States has introduced the concept of implementing a "green port policy", which aims to reduce the port's impact on the environment and enhance its sustainable development capabilities. Subsequently, the Los Angeles Port also launched a "green terminal" plan, which adopts environmentally friendly technologies and measures to reduce port pollution emissions. The New York-New Jersey Port has incorporated the environmental protection concept into its "Port Environmental Management System" to ensure that port operations meet environmental protection standards. The Houston Port Authority has also injected environmental protection concepts into the entire organization, promoting green development of the port. The Rotterdam Port in the Netherlands has proposed a policy of "clean and environmentally friendly ports" by promoting the use of clean energy and reducing emissions of pollutants to improve the environmental friendliness of the port. The Tokyo Port in Japan has set a goal of "efforts to restore the natural ecology of the coastline" by protecting and restoring the marine ecological environment to enhance the port's sustainable development capabilities. Since China joined the WTO in 2001, it has also actively promoted the transformation of port development methods to reduce the adverse effects of ports on the environment. Especially in the context of the implementation of the "double carbon" strategy, green competitiveness has become an important indicator for evaluating the sustainable development capabilities of ports and has become a new focus in port development research. Chinese ports are gradually paying attention to greening and low-carbonization, promoting the harmonious development of ports and environmental protection.

On the other hand, the competitiveness of a port is inseparable from the development of the hinterland economy. The port and its hinterland form a complex spatial area where various elements influence each other. The competitiveness of the port mainly affects the economic vitality and industry development prospects of the hinterland through channels such as logistics transport relationships, geographical location relationships, and industrial relationships. In turn, the product quality, pricing level, and production efficiency of the hinterland economy will also affect the competitiveness of the port. Specifically, ports mainly inject momentum into the economic development of port cities through industrial associative effects, investment multiplier effects, and agglomeration effects, driving the development of the hinterland economy outward. Industrial associative effects refer to the ability of port development to drive the development of related industries, forming a collaborative effect along the industrial chain. Investment multiplier effects refer to the ability of port development to attract more investment, thus promoting the growth of the hinterland economy. Agglomeration effects refer to the ability of port development to attract more population and industries, forming economic clusters that further promote the development of the hinterland economy.

Based on the above perspectives, this article explores the interaction between green competitiveness and the hinterland economy by enhancing the green competitiveness of ports to

promote the sustainable development of the hinterland economy. The main contents include: first, establishing a green competitiveness evaluation index system to measure the greening status of Chinese ports; second, quantitatively measuring the hinterland economy of ports to understand its development; third, constructing a coordination model to study the correlation between green competitiveness and the hinterland economy. By adjusting different variables, the impact of various green variables on the port's hinterland economy can be studied.

In summary, this article creatively integrates the concept of green development into the research on ports and their hinterland economies, providing a reference for the green development of ports under the global emission reduction background. By establishing a green competitiveness evaluation index system, we can measure the greening status of Chinese ports and provide guidance for their green development. By quantitatively measuring the hinterland economy of ports, we can better understand the development status of the hinterland economy and provide a basis for formulating relevant policies. By constructing a coordination model, we can study the correlation between green competitiveness and the hinterland economy, providing a reference for promoting the harmonious development of ports and the hinterland economy.

## 2 Literature review

The study of the relationship between port competitiveness and the hinterland economy offers a crucial perspective on the pivotal role of ports within the global trade and supply chain framework.

In the realm of port competitiveness, academic inquiry primarily focuses on assessing and enhancing the competitive edge of ports, which includes the identification and quantification of factors that influence their competitiveness [3]. Key factors such as port services, hinterland conditions, availability, logistics costs, convenience, and connectivity are recognized as significantly impacting port competitiveness. Traditional assessments of port competitiveness have typically centered on hardware aspects such as infrastructure, service quality, and location. However, with the evolution of global trade patterns and the widespread adoption of sustainable development principles, the evaluation of port competitiveness has broadened to encompass environmental sustainability, technological innovation, service quality, and green practices. For instance, Wu and Yang (2020) explored the role of information technology in enhancing port operational efficiency and service quality [4]. The innovation in port technology, such as smart ports and digital operations, has also been demonstrated to significantly improve port efficiency and attract shipping companies [5]. Additionally, the soft power of ports, including management expertise, service quality, and collaboration with stakeholders in the supply chain, has emerged as a critical indicator of port competitiveness [6]. The interplay between port efficiency and performance is also a subject of extensive research, with evidence indicating that competition can enhance shipping networks and trade environments, thereby bolstering port efficiency and performance [7]. Moreover, the selection of ports is a distinct area of research within port competitiveness, involving a comprehensive evaluation by cargo owners and carriers. Factors such as attractiveness, hinterland connectivity, and other variables influence the port selection process [8]. Furthermore, port governance and policies may evolve in response to competition, and while technological progress is vital for enhancing port competitiveness, breakthroughs in other areas are equally essential to sustain long-term competitive advantages [9].

Regarding green port performance, Kattara and Notteboom (2018) underscored the significance of green port practices in bolstering port competitiveness [10]. Munim and Saeed (2019)

highlighted the sustainability trend in port research, noting that green management and environmentally friendly operations are crucial for enhancing port competitiveness [11]. Zhang et al. (2024) discussed environmental performance indicators for green port development, considering factors such as energy efficiency, the level of technology application, industrial structure adjustment, and environmental protection capacity [12]. Specifically, Rocha et al. (2018) found an overall improvement in environmental sustainability by assessing environmental performance indicators of 30 ports [13]. Additionally, Puig et al. (2020) analyzed 90 European ports and found that these ports performed well in compliance with environmental regulations and the implementation of management policies [14].

In the context of the relationship between ports and the hinterland economy, ports are not only nodes for the distribution of goods but also serve as vital engines for regional economic growth. The economic interaction between ports and their hinterlands is significant for enhancing port competitiveness. The deep integration of ports and the hinterland economy helps to form an economic circle around the port, promoting balanced regional economic development [15]. Factors such as logistics networks, industrial collaboration, and market access conditions between ports and the hinterland economy collectively impact the economic vitality of ports and regional economic development. For instance, by offering specialized services and facilities to attract more manufacturing and logistics companies to establish bases in the hinterland, employment and economic growth can be stimulated [5]. Lirn et al. (2013) highlighted the significant role of a multimodal transportation network between ports and their hinterlands in reducing logistics costs and enhancing transport efficiency [16]. As a means to enhance port competitiveness, dry ports in the hinterland can bolster port competitiveness through cooperation [17]. The integration of ports with inland transportation is crucial for improving the operational efficiency and competitiveness of dry ports [5]. Moreover, the service capabilities of ports in supporting the economy of the hinterland, such as offering customized logistics solutions and promoting trade facilitation measures, are an integral part of port competitiveness [18].

In terms of green development of ports and the hinterland economy, the research by Lee et al. (2021) emphasized the necessity of prioritizing environmental justice in formulating port hinterland policies, particularly the importance of procedural justice [19], as well as the urgency of resident participation and construction of environmental protection infrastructure. The review of global port strategies by Aregall et al. (2018) revealed the insufficient measures to improve the environmental conditions in port hinterlands, highlighting the need to extend green strategies to the regional level [20]. The study by Cong et al. (2020) further confirmed the crucial role of ports in promoting economic development in port cities, emphasizing the importance of green development of ports and their hinterlands in achieving both economic and environmental benefits [21]. These studies collectively point to the complexity and multidimensionality of policy-making for green development of ports and their hinterlands, as well as the future direction for achieving harmonious coexistence and sustainable development between ports and their hinterlands.

Above all, existing research primarily focuses on the evaluation of port competitiveness and its direct impact on the hinterland economy, with relatively less attention given to the interactive relationship between port green competitiveness and the hinterland economy. Literature suggests that the enhancement of port competitiveness can foster the development of the hinterland economy, but the role of green competitiveness in this process has not yet been fully explored. This paper aims to fill this research gap by constructing an evaluation index system for port green competitiveness, combining it with empirical analysis to explore the impact of green competitiveness on the hinterland economy and its underlying mechanisms.

# 3 Evaluation of port green competitiveness and hinterland economic system

## 3.1 Evaluation of port green competitiveness

**3.1.1 Indicator selection.** In this paper, three major categories of indicators that reflect the efficiency of green operations and environmental friendliness of ports are selected based on production theory and principles of environmental economics (Table 1). These indicators include inputs, outputs, and undesirable outputs, which differentiate various factors and outcomes in the production process. They encompass aspects such as energy consumption, pollution control, and the ability to provide green services at ports. These indicators cover not only port operations but also environmental protection and social responsibility, enabling a comprehensive assessment of the economic benefits and environmental performance of ports. The specific indicators are as follows:

**(1) Input Indicators**

In the port green competitiveness indicator system, input indicators are crucial parameters for measuring the level of port infrastructure. This paper selects three indicators: the number of berths, the number of ten-thousand-ton berths, and the length of berths. These indicators together provide a fundamental evaluation of port handling capacity. The number of berths directly relates to the number of vessels that can be accommodated at the port, which is an important reflection of the scale of port infrastructure. The more berths a port has, the stronger its theoretical cargo handling capacity. Ten-thousand-ton berths specifically refer to the number of berths capable of accommodating vessels of 10,000 tons or more. This indicator reflects the port's ability to receive large vessels and is an important sign of modernization and international competitiveness. The length of berths refers to the total length of all berths. It is not only related to the port's cargo handling capacity but also affects the efficiency and safety of vessel berthing. Data for these input indicators can typically be obtained from port authorities, transportation departments, or relevant industry reports. They provide a quantitative standard for the infrastructure dimension in evaluating the green competitiveness of ports.

**(2) Output Indicators**

Output indicators are key to measuring the operational efficiency of ports and include cargo throughput, container throughput, foreign trade cargo throughput, and foreign trade container throughput. These indicators reflect the production capacity of the port and its position in international trade. Cargo throughput refers to the total amount of cargo handled

**Table 1. Indicators for evaluating port green competitiveness.**

| Category | Indicators | Remark |
|---|---|---|
| Input Indicators | Number of Berths | [22] |
| | Number of Berths for Mega Ships (over 10,000 DWT) | [23] |
| | Length of Berths | [24] |
| Output Indicators | Cargo Throughput (Ton) | [25] |
| | Container Throughput (TEU) | [26] |
| | Foreign Trade Cargo Throughput (Ton) | [27] |
| | Foreign Trade Container Throughput (TEU) | [28] |
| Undesirable Output Indicators | Industrial Wastewater Emission (Ton) | [29] |
| | Industrial Dust Emission (Ton) | [30] |
| | Industrial Sulfur Dioxide Emission (Ton) | [31, 32] |

through the port during the statistical period. This indicator is the basic measure of the port's production scale and operational capability. Among them, containers, as the main transport unit in modern international trade, have their throughput as an important indicator of the modernization level and service quality of ports. Foreign trade cargo throughput and foreign trade container throughput focus on measuring the throughput capacity of ports in international trade. The level of these indicators directly affects the international competitiveness and status of the port. Data for these output indicators usually come from the port's operational records and official statistical reports. They provide a quantitative basis for evaluating the green competitiveness of ports from the perspective of production efficiency.

### (3) Undesirable Output Indicators

Undesirable output indicators focus on the environmental impact of port operations, including industrial wastewater emissions, industrial dust emissions, and industrial sulfur dioxide emissions. These indicators reflect the environmental responsibility and challenges of ports. Industrial wastewater emissions refer to the amount of industrial wastewater emitted by the port during operations. This indicator relates to the level of pollution to surrounding water bodies. Industrial dust emissions refer to the total amount of dust emitted during port operations. Dust is one of the main components of air pollution, and its emissions directly affect air quality. In addition, sulfur dioxide is one of the main gases causing acid rain, and its emissions reflect the impact of port operations on the atmospheric environment. Data for these undesirable output indicators typically require monitoring and reporting by environmental protection departments. They provide quantitative indicators for evaluating the green competitiveness of ports from the perspective of environmental protection. Through the monitoring and improvement of these indicators, ports can better achieve their sustainable development goals.

**3.1.2 Super-efficiency Slacks-Based Measure (SBM).** The Super-Efficiency SBM is a sophisticated approach derived from Data Envelopment Analysis (DEA) designed to assess the efficiency of Decision Making Units (DMUs) [33].Ports are intricate systems with multiple inputs and outputs, including such factors as the number and capacity of berths, as well as cargo and container throughput, among others. The Super-Efficiency SBM's capacity to simultaneously analyze these diverse inputs and outputs allows for a more precise measurement of port production efficiency [34]. Additionally, the model excels in addressing unwanted outputs, such as industrial wastewater and air pollution emissions, which are unavoidable byproducts of port operations. By incorporating these negative externalities into its evaluation, the model offers a more holistic assessment of a port's green competitiveness. Furthermore, the Super-Efficiency SBM can distinguish between the economies of scale across different ports, providing valuable insights into their operational peculiarities and competitive advantages. It also identifies specific areas for improvement by suggesting which inputs might be reduced, which outputs could be increased, and how to manage unwanted outputs, aiding port managers in developing more strategic and effective approaches. Consequently, this paper employs the Super-Efficiency SBM to gauge the green competitiveness of ports, delivering a more refined evaluation. Comparing the green efficiency of various ports facilitates the identification of their relative strengths and weaknesses in pursuit of sustainable development. The model's application involves the following steps:

1. Define Decision Units
   Consider $n$ decision units ($j = 1,2,\cdots,n$), each characterized by $m$ types of input indicators ($i = 1,2,\cdots,m$) and $q$ types of output indicators ($r = 1,2,\cdots,q$).

2. Construct Input and Output Matrices
   Create an $n * m$ input matrix $X$, where each row represents the values of m inputs for a

DMU, and an $n * q$ output matrix $Y$, where each row signifies the values of q outputs for a DMU, with $X_{jm}$ denoting the amount of the $m^{th}$ input for the $j^{th}$ DMU, and $X_{jq}$ representing the amount of the qth output.

3. Calculate Efficiency Values

Adhering to [35], the Super-Efficiency SBM's formula is as follows:

$$min\rho = \frac{1 + \frac{1}{m}\sum_{i=1}^{m}\frac{s_i^-}{x_{jm}}}{1 - \frac{1}{q}\sum_{i=1}^{q}\frac{s_i^+}{y_{jq}}}, s.t. \begin{cases} \sum_{j=1,j\neq k}^{n} x_j\lambda_j - s_i^- \leq x_k \\ \sum_{j=1,j\neq k}^{n} y_j\lambda_j + s_i^+ \geq y_k \\ \lambda, s^-, s^+ \geq 0 \\ i = 1, 2, \cdots, m; r = 1, 2, \cdots, q; j = 1, 2, \cdots, n(j \neq k) \end{cases} \quad (1)$$

Here, $\rho$ denotes the efficiency value, with $s^-$ and $s^+$ being inversely related; a higher value of these slack variables corresponds to a lower efficiency value. An efficiency value of $\rho \geq 1$ indicates an effective DMU, otherwise, it is considered inefficient. $\lambda$ represents the weight vector, while $s_i^-$ and $s_r^+$ are the input and output slack variables, respectively. The constraint $j \neq k$ signifies that when assessing the efficiency of DMU $j$, the comparison is solely based on the linear combination of this DMU with all other DMUs in the sample, excluding DMU $j$ itself.

**3.1.3 Result analysis.** Drawing from the "guidelines on developing comprehensive transport network" issued by the the State Council of China, which delineates the principal coastal ports and considering the significance of these ports as well as the accessibility of data, this paper will primarily focus on the development of 25 major coastal ports and conduct empirical research. Geographically, these ports are distributed across different port groups as outlined by the Chinese government's strategic planning. The coastal region has coalesced into five major port groups, arrayed from north to south around the Bohai Rim, Yangtze River Delta, Southeast Coastal, Pearl River Delta, and Southwest Coastal areas (see Table 2 for details). To inform our analysis, we have utilized the "China Port Yearbook" to compile panel data on the green competitiveness indicators of these ports, spanning the years 2003 to 2021.

Fig 1 delineates the trajectories of green competitiveness for the 25 specified ports. Collectively, these ports demonstrate an overall increasing trend in green competitiveness; however, the rates of growth vary significantly among individual ports. Shanghai, Ningbo-zhoushan, and Shenzhen are at the forefront of this green competitiveness growth, which is likely due to their proactive engagement with environmental protection policies, technological innovation, and capital investment. Furthermore, the ports within different port groups exhibit varying levels of green competitiveness. Data analysis reveals marked disparities in green competitiveness among these port groups, with the Yangtze River Delta group standing out for its robust economic growth, significant government investment in green technology and environmental protection, and advanced management practices. By comparison, although the Bohai Rim group shows some green competitiveness, it may confront more substantial environmental protection challenges, given the concentration of heavy industry in the area. The Southeast Coastal group, including ports like Fuzhou and Xiamen, despite rapid economic expansion, might not have allocated as much resources to green technology and environmental protection as the Yangtze River Delta. The Pearl River Delta group, centered on Guangzhou and Shenzhen Ports, as a leader in reform and opening up, sees its green competitiveness fueled by economic dynamism, yet it still lags behind the Yangtze River Delta. The Southwest Coastal group has a later start in economic development and may face relative shortages in funding and technological investment, leading to its lower green competitiveness. In summary, the green

**Table 2. Main coastal ports in China.**

| Port Group | Port |
|---|---|
| Bohai Rim Port Group | Dalian |
| | Tianjin |
| | Qinhuangdao |
| | Tangshan |
| | Yingkou |
| | Qingdao |
| | Rizhao |
| | Yantai |
| | Lianyungang |
| Yangtze River Delta Port Group | Shanghai |
| | Nanjing |
| | Nantong |
| | Zhenjiang |
| | Suzhou |
| | Ningbo-zhoushan |
| Southeast Coastal Port Group | Wenzhou |
| | Xiamen |
| | Fuzhou |
| Pearl River Delta Port Group | Guangzhou |
| | Shenzhen |
| | Zhuhai |
| | Shantou |
| Southwestern Coastal Port Group | Fangcheng |
| | Haikou |
| | Zhanjiang |

competitiveness of ports is shaped by a multitude of factors, including economic power, geographical location, policy context, capital investment, and technology adoption. The discrepancies among the port groups reflect their distinct circumstances and the challenges they face in these areas.

## 3.2 Evaluation of hinterland economic system

**3.2.1 Indicator selection.** The factors influencing the economy of a port's hinterland are numerous and intricate, necessitating a holistic consideration of various elements. These include geographical positioning and infrastructure, economic development, international trade, industrial composition, labor market dynamics, and the policy environment. This paper constructs a comprehensive index framework for assessing the economic system of a port's hinterland, referred to as the port hinterland economic system, which aims to evaluate the composite development of this economic area. The assessment of the port hinterland economic system encompasses three key dimensions: the Total Hinterland Economy (HlT), the Structure of the Hinterland Economy (HlS), and the Quality of the Hinterland Economy (HlQ). At each of these levels, distinct sub-indicators are chosen to quantify the performance. (See Table 3 in details) HIT refers to the overall economic size and activity of the region that is served by the port, which encompasses the gross domestic product (GDP), total social fixed asset investment, total retail sales of consumer goods, the total import and export trade, and

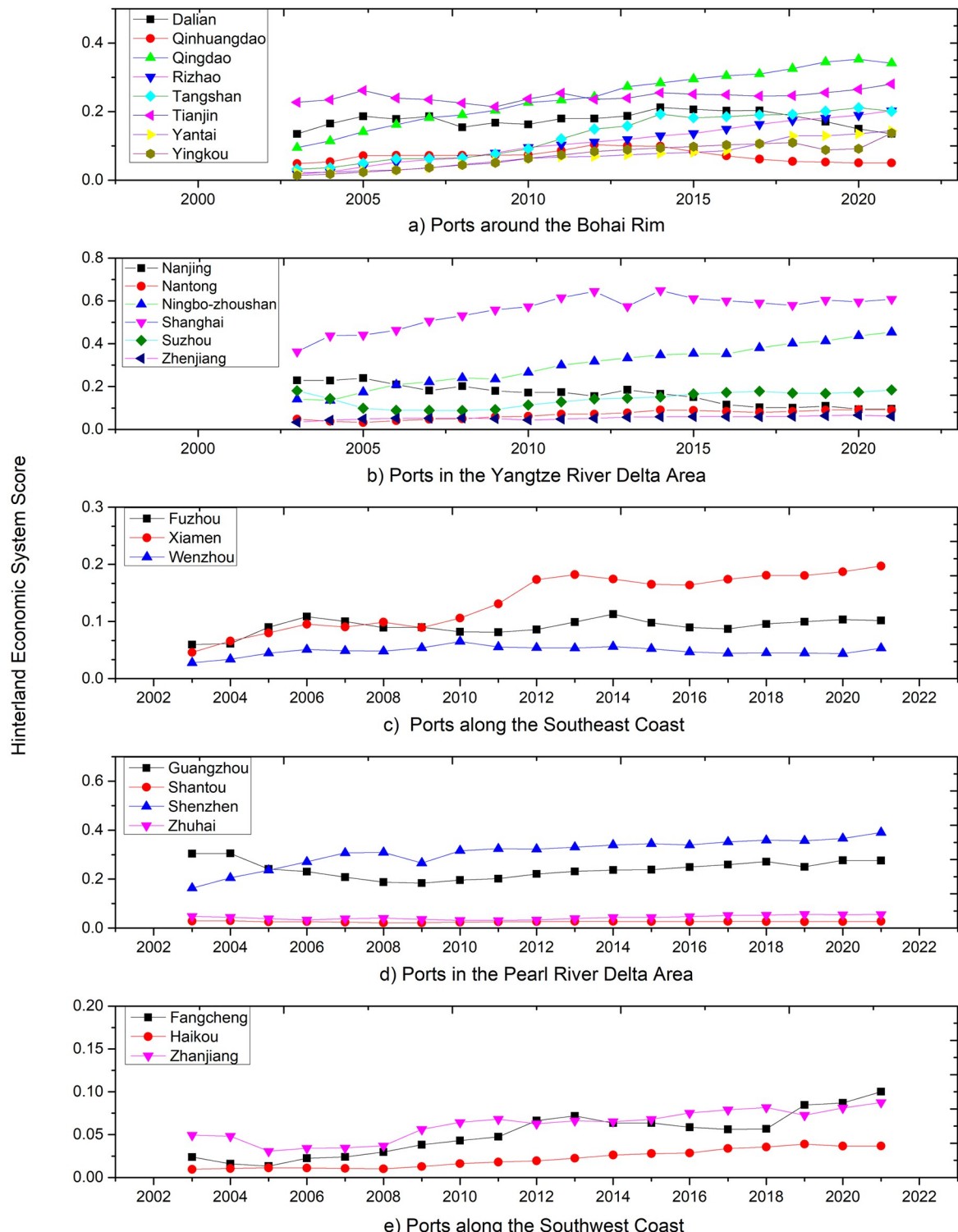

**Fig 1. Green competitiveness of main coastal ports in China.**

**Table 3. Evaluation index system for port hinterland economy.**

| Category | Indicators |
|---|---|
| Total Hinterland Economy (HlT) | Gross Domestic Product |
| | Total Social Fixed Asset Investment |
| | Total Retail Sales of Consumer Goods |
| | Total Import and Export Trade |
| | Foreign Direct Investment |
| Structure of the Hinterland Economy (HlS) | Industrial Output Value as a Percentage of GDP |
| | Tertiary Industry Output Value as a Percentage of GDP |
| | Fixed Asset Investment as a Percentage of GDP |
| Quality of the Hinterland Economy (HlQ) | Per Capita Disposable Income |
| | Tertiary Industry Value Added Index |
| | Per Capita Gross Regional Product |
| | Total Social Labor Productivity |

Source: Statistical Yearbooks of Each City & China Stock Market Accounting Research Database

foreign direct investment. It provides an understanding of the economic scale and importance of the hinterland to the port's operations and its potential for generating cargo and trade. HIS examines the composition and diversity of economic sectors within the hinterland, which includes the proportion of industrial output value in GDP, the share of tertiary industry output value in GDP, and the percentage of fixed asset investment in GDP. A well-structured hinterland economy is typically more resilient and capable of supporting sustainable growth for the port. HIQ assesses the efficiency, productivity, and competitiveness of the hinterland's economic activities. HlQ takes into account factors such as per capita disposable income, the index of tertiary industry added value, per capita gross regional product, and the societal labour productivity. A high-quality hinterland economy can enhance the port's ability to attract and retain business, as well as improve the overall performance and competitiveness of the port-hinterland system.

**3.2.2 Entropy weight TOPSIS.** The Entropy Weight TOPSIS Method is a sophisticated multi-attribute decision-making analysis technique that integrates the Entropy Weight method and the TOPSIS method. This method is widely employed for intra-group comprehensive evaluations due to its ability to harness the full potential of the original data, providing results that vividly depict the distinctions between various evaluation options. Application of the Entropy Weight TOPSIS method enables a thorough and objective assessment of the economic development within a port's hinterland, thereby offering a scientific foundation for policy formulation, resource allocation, and regional economic progress. The method's calculation steps are outlined as follows:

1. Creation of a Normalized Matrix in the Same Direction
   Presuming the original decision matrix is denoted by $X$:

$$X = \begin{bmatrix} x_{11} & \cdots & x_{1m} \\ \cdots & \cdots & \cdots \\ x_{n1} & \cdots & x_{nm} \end{bmatrix} \tag{2}$$

The normalized decision matrix $X = \left[ x_{ij} \right]_{m \times n} (i = 1, 2, \cdots, n; j = 1, 2, \cdots, m)$ is constructed, where $x_{ij}$ represents the performance of the $i$ option on the $j^{th}$ indicator.

2. Data Standardization
To neutralize the effects of varying indicator scales, the decision matrix is standardized using the maximum-minimum method:

$$x_{ij}' = \frac{x_{ij} - \min_{j} x_{ij}}{\max_{j} x_{ij} - \min_{j} x_{ij}} \tag{3}$$

$x_{ij}'$ signifies the standardized value.

3. Weight Determination Weights for each indicator are computed using the Entropy Weight method. Commencing with the entropy calculation for the $j^{th}$ indicator:

$$H_j = -k \sum_{i=1}^{m} p_{ij} \, ln p_{ij} \tag{4}$$

Here, $p_{ij} = \frac{x_{ij}'}{\sum_{i=1}^{m} x_{ij}'}$, $k = 1/lnm$ serves as the constant. Subsequently, the weight for the $j^{th}$ indicator is derived:

$$w_j = \left( 1 - H_j \right) / \sum_{j=1}^{n} \left( 1 - H_j \right) \tag{5}$$

4. Establishment of the Weighted Decision Matrix
The standardized decision matrix is then multiplied by the weights to obtain the weighted decision matrix:

$$Z^* = \left( z_{ij} \right)_{n \times m} (i = 1, 2, \cdots, n; j = 1, 2, \cdots, m) \tag{6}$$

Where $z_{ij} = w_j x_{ij}'$

5. Identification of Positive and Negative Ideal Solutions
The positive ideal solution encompasses the maximum values across all indicators:

$$Z^+ = (max\{z_{11}, z_{21}, \cdots, z_{n1}\}, \cdots, max\{z_{1m}, z_{2m}, \cdots, z_{nm}\})$$

Conversely, the negative ideal solution involves the minimum values across all indicators:

$$Z^- = (min\{z_{11}, z_{21}, \cdots, z_{n1}\}, \cdots, min\{z_{1m}, z_{2m}, \cdots, z_{nm}\})$$

6. Computation of Distance to Ideal Solutions
The Euclidean distance between the $i^{th}$ evaluation object and the maximum value $Z^+$ is calculated as $D_i^+$:

$$D_i^+ = \sqrt{\sum_{j=1}^{m} \left( Z_i^+ - z_{ij} \right)^2} \tag{7}$$

Similarly, the distance to the minimum value $Z^-$ is computed as $D_i^-$:

$$D_i^- = \sqrt{\sum_{j=1}^{m} \left( Z_i^- - z_{ij} \right)^2} \tag{8}$$

$D_i^+$ represents the distance to the positive ideal solution (the most satisfactory solution across all indicators), while $D_i^-$ signifies the distance to the negative ideal solution (the least satisfactory solution across all indicators).

7. Calculation of Relative Closeness

The unnormalized score for the $i^{th}$ evaluation object is determined by:

$$S_i = \frac{D_i^-}{D_i^+ + D_i^-} \tag{9}$$

8. Subsequently, this score is normalized to reflect the relative closeness of the alternative to the positive ideal solution, represented by:

$$C = \frac{S_i}{\sum_{i=1}^{n} S_i}, C \in [0, 1] \tag{10}$$

A higher $C$ value indicates a superior option.

### 3.2.3 Result analysis

Fig 2 tracks the scores of the hinterland economic systems for China's major coastal ports from 2003 to 2021. It encompasses ports from the Bohai Bay, Yangtze River Delta, Pearl River Delta, Southwestern, and Southeastern coastal regions. Each port is color-coded, with the trajectory of their respective hinterland economic system scores over time clearly mapped out. Across all regions, there is a consistent upward trend in port scores, signifying significant growth over the past 15 years. This upwards trajectory mirrors the rapid expansion of China's economy, as well as the surge in foreign trade and shipping demands. During this period, China's port sector has undergone transformative changes, including infrastructure build-outs, operational efficiency enhancements, and the integration of modern and automated technologies. These advancements have not only boosted the cargo handling capacity of the ports but also elevated China's stature in the global trade landscape.

Regionally, Fig 2a) details the evolution of the Bohai Rim port group's hinterland economy from 2003 to 2021. Tianjin stands out as the leader in hinterland economic development, significantly outpacing its Bohai Rim counterparts. Dalian exhibited robust development between 2000 and 2014, but experienced a sharp downturn post-2014, with its hinterland economic development trailing behind Yantai and Tangshan. Yantai and Tangshan exhibit similar patterns in hinterland economic development. Rizhao Port, Qinhuangdao, and Yingkou follow a more aligned trajectory, with consistent changes over time.

Fig 2b) showcases the development trend of the Yangtze River Delta port group's hinterland economy from 2003 to 2021. Shanghai emerges as the clear leader in hinterland economic development, far surpassing other ports. Suzhou follows, with its hinterland economic development value roughly equivalent to half of Shanghai. The trends and magnitudes of Nanjing

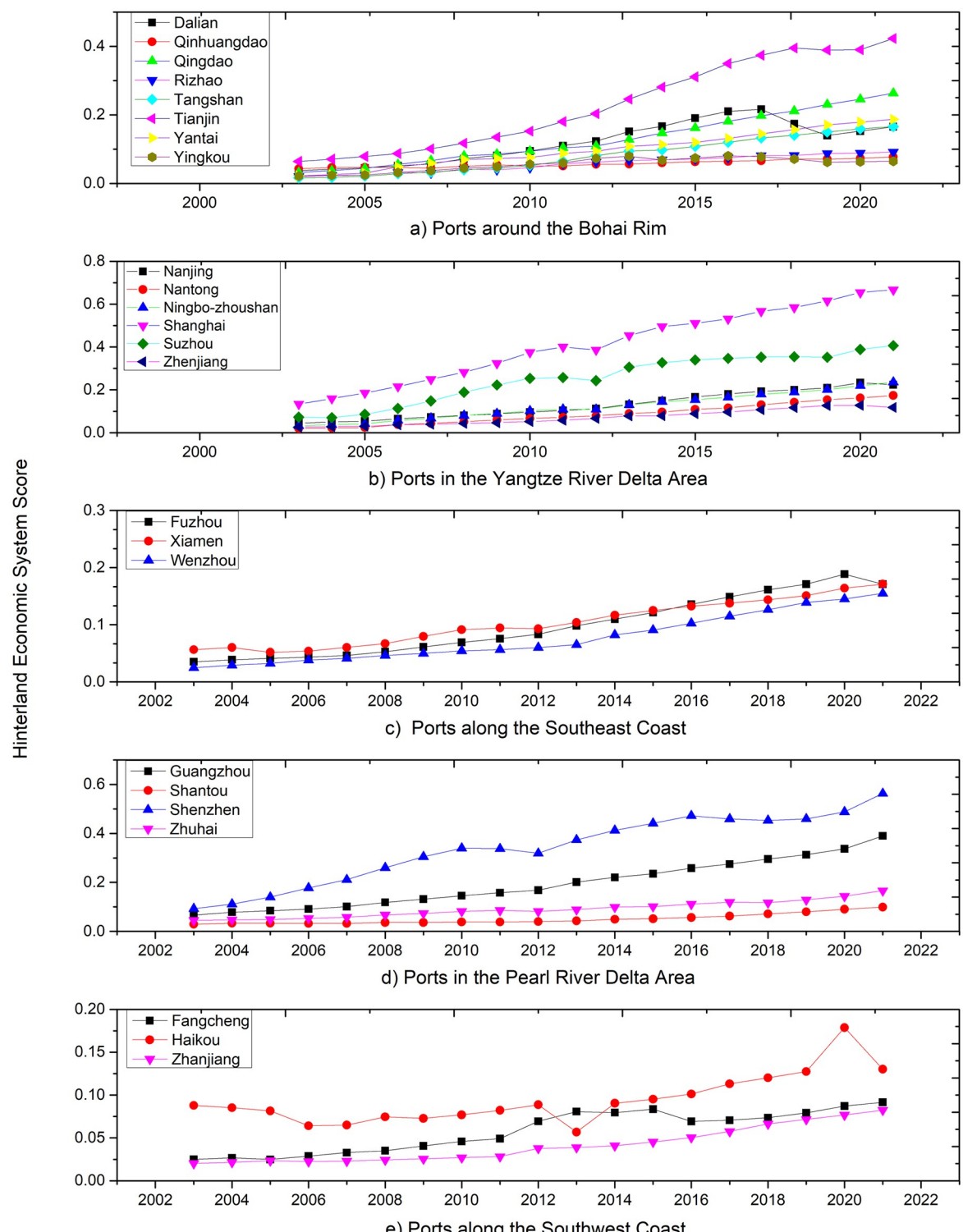

**Fig 2. Evolution of the hinterland economic systems at major coastal ports in China.**

and Ningbo Zhoushan's hinterland economic development are relatively consistent, while Zhenjiang and Lianyungang exhibit similar trends in hinterland economic development.

Fig 2c) illustrates the changes in the Southeastern coastal port group's hinterland economy from 2003 to 2021. Xiamen and Fuzhou exhibit parallel changes, with Xiamen's hinterland economic development being higher than Fuzhou's in 2003–2013 and 2018–2021, yet lower than Fuzhou's in 2014–2017.

Fig 2d) documents the changes in the Pearl River Delta port group's hinterland economy from 2003 to 2021. The hierarchy of hinterland economic development is evident, with Shenzhen Port at the forefront, followed by Guangzhou, Zhuhai, and Shantou. The Pearl River Delta port group's hinterland economic development levels vary significantly, with an anomaly in 2021 where Shenzhen's hinterland economic development index fell below that of Guangzhou. In other years, the development trend adheres to the established pattern.

Fig 2e) presents the changes in the Southwestern coastal port group's hinterland economy from 2003 to 2021. Haikou stands out with the highest level of hinterland economic development, far exceeding the other two ports, except for a brief period in 2010 when its level was lower than Fangcheng. Fangcheng's hinterland economic development level is higher than Zhanjiang, except in 2020 when it was lower. Zhanjiang rounds out the trio with the lowest level of hinterland economic development among the three ports, but it shows a trend of annual growth.

Overall, the changes in the scores of the economic systems of the coastal ports in China reflect the imbalance in regional economic development and the competitive and cooperative relationships between ports. Large ports perform more prominently in the scores of the economic systems due to their scale effects and comprehensive service capabilities. Small and medium-sized ports, on the other hand, need to look for new growth points in areas such as specialized services and regional cooperation.

## 4 The impact of port green competitiveness on the hinterland economic system

### 4.1 Model specification

The multifaceted and profound impact of enhancing port competitiveness on the hinterland economic system necessitates a nuanced analysis through sophisticated empirical models. Drawing on economic theory, the following benchmark regression model is established:

$$Hinterland_{i,t} = \alpha_0 + \alpha_1 Port_{i,t} + \alpha_2 Controls_{i,t} + \omega_t + \mu_i + \varepsilon_{i,t} \tag{11}$$

Where $Hinterland_{i,t}$ denotes the development level of the economic system of the hinterland for port $i$ in year $t$. $Port_{i,t}$ represents the competitiveness level of port $i$ in year $t$. $\alpha_0$ stands for the intercept, $\alpha_1$ is the coefficient indicating the impact of port competitiveness on the hinterland economic system's development level. $Control_{i,t}$ encompasses a set of control variables selected for this paper, $\omega_t$, $\mu_i$ represent time-specific and port-specific fixed effects, respectively. $\varepsilon_{i,t}$ captures the random error term.

The dependent variable in this model is the hinterland economic system (*Hinterland*), with its construction and measurement detailed in *section 3.2*. Similarly, the total hinterland economy (*HlT*), the structure of the hinterland economy (*HlS*), and the quality of the hinterland economy (*HlQ*) are measured using the same approach.

The independent variable in this model is the port green competitiveness (PG), with its construction and measurement described in section 3.1.

In addition, following the insights of previous scholarly research, the following control variables are included: (1) Population size (*Popula*), quantified by the natural logarithm of the

average annual population in the sample city; (2) Science and technology expenditure (*Science*), measured through local science and technology fiscal outlays; (3) Education level (*Edu*), captured by the natural logarithm of educational expenditures in the sample city; (4) Marketization index (*Market*), calculated using the marketization index compiled by [36], which encompasses sub-indicators such as the relationship between government and market, non-state-owned economic development, product market development, factor market development, intermediary organization development, and the rule of law; (5) Local fiscal budgetary revenue (*Revenue*); (6) Government intervention (*Intervent*), gauged by the proportion of local fiscal budgetary revenue to GDP; (7) Economic development scale (*Gdp*), measured by the per capita gross regional product; (8) Foreign investment intensity (*Fdi*), determined by the ratio of foreign direct investment to regional GDP; (9) Freight volume (*Road*), quantified by road freight volume.

Employing the port scope previously defined, this section conducts empirical research on the panel data of the hinterland economic development level for 25 sample ports, utilizing the descriptive statistics of the main variables. Green data statistics commenced in 2003. The data on the hinterland economy are sourced from the statistics bureaus of each city and the "City Statistical Yearbook"; other control variables are obtained from the "City Statistical Yearbook". Table 4 presents the descriptive statistics of the main variables. The minimum and maximum values for port green competitiveness are 0.001 and 1.090, respectively; for the hinterland economic system, they are 0.016 and 0.721; for the total hinterland economy, 0.000 and 0.726; for the structure of the hinterland economy, 0.055 and 0.821; and for the quality of the hinterland economy, 0.017 and 0.727. Notable disparities exist among sample cities in science and technology expenditure, foreign investment intensity, and marketization index, which may significantly influence port green competitiveness. The discrepancies in population size, education level, local fiscal budgetary revenue, government intervention, economic development scale, and freight volume are relatively small, suggesting a more balanced impact and performance across sample cities. Concurrently, the presence of negative values in economic development scale suggests that the economic development level of some cities falls below the average level.

**Table 4. Descriptive statistics of main variables.**

| Variable | Observations | Mean | Standard Deviation | Minimum | Median | Maximum |
|---|---|---|---|---|---|---|
| PG | 494 | 0.090 | 0.194 | 0.001 | 0.032 | 1.090 |
| Hinterland | 572 | 0.136 | 0.124 | 0.016 | 0.089 | 0.721 |
| HIT | 572 | 0.118 | 0.142 | 0.000 | 0.061 | 0.726 |
| HIS | 572 | 0.355 | 0.117 | 0.055 | 0.361 | 0.821 |
| HIQ | 572 | 0.118 | 0.114 | 0.017 | 0.081 | 0.727 |
| Popula | 572 | 6.040 | 0.772 | 4.049 | 6.359 | 7.308 |
| Science | 572 | 10.163 | 2.477 | 3.045 | 10.187 | 15.529 |
| Edu | 572 | 12.467 | 1.586 | 8.194 | 12.584 | 16.157 |
| Market | 572 | 9.940 | 3.556 | 1.843 | 9.900 | 18.974 |
| Revenue | 572 | 14.425 | 1.554 | 10.392 | 14.404 | 18.104 |
| Intervent | 572 | 0.084 | 0.033 | 0.028 | 0.079 | 0.214 |
| Gdp | 572 | 7.763 | 1.327 | -1.168 | 7.740 | 10.674 |
| Fdi | 572 | 11.531 | 1.695 | 6.438 | 11.755 | 15.804 |
| Road | 572 | 9.132 | 0.996 | 5.165 | 9.226 | 11.523 |

## 4.2 Baseline regression analysis

This section investigates the influence of port green competitiveness on the hinterland economic system, ensuring the reliability of the findings by incrementally incorporating controlled variables and fixed effects. The regression outcomes are presented in Table 5. In the first column, no control variables or fixed effects are employed, and the estimated coefficient for green competitiveness is 0.0295, which is statistically significant at the 1% level. This suggests a positive correlation between the green competitiveness of ports and the hinterland economic system. The second column adds additional control variables, while the third column introduces both year and individual fixed effects without any abnormal changes in the coefficients. The results indicate that enhancing port green competitiveness can substantially refine the hinterland economic system.

Table 5 shows that the parameter for EDU is insignificant in the second model but significant in the third model, whereas the parameter for FDI shows the opposite pattern. These differences can be attributed to variations in model specifications and the inclusion of different

**Table 5. Regression results of the impact of port green competitiveness on the hinterland economic system.**

| Variable | (1) | (2) | (3) |
|---|---|---|---|
| | Hinterland | Hinterland | Hinterland |
| PG | 0.0295*** | 0.0100** | 0.0298* |
| | (0.011) | (0.005) | (0.016) |
| Popula | | 0.0458** | -0.0461*** |
| | | (0.021) | (0.006) |
| Science | | 0.0125*** | -0.0155*** |
| | | (0.005) | (0.004) |
| Edu | | -0.0020 | 0.0286*** |
| | | (0.012) | (0.007) |
| Market | | -0.0133*** | -0.0026** |
| | | (0.004) | (0.001) |
| Revenue | | -0.2621*** | -0.2775*** |
| | | (0.031) | (0.033) |
| Intervent | | 2.6530*** | 3.9848*** |
| | | (0.352) | (0.370) |
| Gdp | | 0.2681*** | 0.3784*** |
| | | (0.033) | (0.035) |
| Fdi | | 0.0067** | 0.0036 |
| | | (0.003) | (0.003) |
| Road | | -0.0223*** | -0.0182*** |
| | | (0.006) | (0.006) |
| Constant | -0.3983*** | 0.0042 | -0.7877*** |
| | (0.026) | (0.025) | (0.049) |
| Year | No | No | Yes |
| Id | No | No | Yes |
| Observations | 494 | 494 | 494 |
| R-squared | 0.625 | 0.797 | 0.929 |

Note: Throughout the tables in this manuscript, asterisks are used to denote the levels of statistical significance for the reported results. Specifically:

"***" indicates statistical significance at the 1% level (p < 0.01),

"**" indicates statistical significance at the 5% level (p < 0.05),

"*" indicates statistical significance at the 10% level (p < 0.10).

control variables, which can affect the statistical significance of the parameters. For example, the inclusion of additional variables in the third model may account for the variance that affects the significance of EDU and FDI differently.

The third model includes a broader set of variables, which may provide a more comprehensive understanding of the factors influencing the hinterland economy. However, it is important to note that the parameter for PG is only statistically significant at the 10% level in this model. This lower level of significance suggests that while the third model offers a more detailed analysis, the evidence for the impact of PG is weaker compared to other variables.

When determining which model to prefer, we consider both the statistical significance of the key variables and the overall explanatory power of the models. While the third model provides more detailed insights, the lower significance level of PG warrants cautious interpretation. We recommend using the third model for a more comprehensive analysis but acknowledging the limitations regarding the significance of the PG parameter.

As shown in Table 6, the regression is conducted separately from three sub-dimensions of the hinterland economic system to examine the impact of port green competitiveness on the total volume, structure, and quality of the hinterland economy. The regression results reveal that when the total volume and quality of the hinterland economy are used as the dependent variables, the estimated coefficients for port green competitiveness are significantly positive at

**Table 6. Regression results of the impact of port green competitiveness on sub-dimensions of the hinterland economic system.**

| Variable | (1) | (2) | (3) |
|---|---|---|---|
| | HIT | HIS | HIQ |
| PG | 0.0091* | 0.0005 | 0.0089* |
| | (0.005) | (0.011) | (0.005) |
| Popula | 0.0902*** | -0.0150 | 0.0173 |
| | (0.024) | (0.026) | (0.020) |
| Science | 0.0137*** | 0.0064 | 0.0073* |
| | (0.005) | (0.006) | (0.004) |
| Edu | 0.0216* | 0.0224 | -0.0473*** |
| | (0.013) | (0.014) | (0.011) |
| Market | -0.0147*** | -0.0183*** | -0.0116*** |
| | (0.004) | (0.004) | (0.003) |
| Revenue | -0.2775*** | 0.0730* | -0.2674*** |
| | (0.035) | (0.039) | (0.030) |
| Intervent | 2.6795*** | -0.5222 | 3.0846*** |
| | (0.392) | (0.431) | (0.331) |
| Gdp | 0.2668*** | -0.1107*** | 0.3020*** |
| | (0.036) | (0.040) | (0.031) |
| Fdi | 0.0100*** | 0.0086** | -0.0004 |
| | (0.003) | (0.003) | (0.003) |
| Road | -0.0315*** | 0.0009 | -0.0200*** |
| | (0.006) | (0.007) | (0.005) |
| Constant | 1.2099*** | 0.0568 | 2.1129*** |
| | (0.302) | (0.332) | (0.255) |
| Year | Yes | Yes | Yes |
| Id | Yes | Yes | Yes |
| Observations | 494 | 494 | 494 |
| R-squared | 0.935 | 0.839 | 0.925 |

the 10% level. However, when the structure of the hinterland economy is the dependent variable, the estimated coefficient is positive but not significant. This suggests that the improvement of port green competitiveness can significantly enhance the total volume and quality of the hinterland economy, whereas its optimizing effect on the structure of the hinterland economy is not significant. In other words, the optimizing effect of the increase in green competitiveness on the hinterland economic system is primarily manifested in the improvement of the total volume and quality of the hinterland economy.

In our analysis, we observed that the significance of some variables' parameters changes between different models. Several factors can explain this variability. Firstly, changes in model specification, such as the inclusion or exclusion of certain control variables, can significantly impact the estimated parameters and their significance. Secondly, differences in sample size and composition across models can lead to variations in parameter estimates. Finally, multicollinearity among the predictors can affect the stability of parameter estimates, causing fluctuations when different sets of variables are included in the models. Understanding these factors is crucial for interpreting the results accurately and for developing robust regression models.

### 4.3 Robust test

**4.3.1 Change of dependent variables.** In the baseline regression mentioned above, the entropy-weighted TOPSIS method was employed to measure the hinterland economic system, the total volume of the hinterland economy, the structure of the hinterland economy, the quality of the hinterland economy, and port green competitiveness. To verify the robustness of the baseline regression results, this section re-measures the dependent variables. Specifically, the entropy-weighted method is used to recalculate the hinterland economic system, the total volume of the hinterland economy, the structure of the hinterland economy, and the quality of the hinterland economy. The remaining variables remain consistent with the baseline regression. The regression results, as shown in Table 7, reveal that after changing the measurement method of the dependent variables, there is no significant difference in the regression coefficients of port green competitiveness, further confirming the robustness of the baseline regression results.

Similarly, dimensional regressions are conducted on the changed key variables to examine the robustness of the research conclusions from the baseline regression regarding the impact of port green competitiveness on the total volume, structure, and quality of the hinterland economy. The regression results, as shown in Table 8, indicate that the estimated coefficients for port green competitiveness are still significantly positive, further suggesting that the improvement of port green competitiveness can significantly enhance the total volume and quality of the hinterland economy and optimize the structure of the hinterland economy.

**4.3.2 Change of core explanatory variable.** In this section, the core explanatory variable is recalculated, which is the green competitiveness calculated using the SBM model, with the remaining variables consistent with the baseline regression. The regression is re-conducted with the changed variables to examine the robustness of the research conclusions from the baseline regression regarding the impact of port green competitiveness on the hinterland economic system. The regression results, as shown in Table 9, show that regardless of whether controlled variables and fixed effects are included, the estimated coefficients for port green competitiveness are still significantly positive, further indicating that the improvement of port green competitiveness can significantly optimize the hinterland economic system.

**Table 7. Regression results of the impact of port green competitiveness on the hinterland economic system (after changing dependent variables).**

| Variable | (1) | (2) | (3) |
|---|---|---|---|
| | Hinterland | Hinterland | Hinterland |
| PG | 0.6309*** | 0.3792*** | 0.4841*** |
| | (0.016) | (0.020) | (0.023) |
| Popula | | -0.0107*** | 0.0948*** |
| | | (0.004) | (0.010) |
| Science | | -0.0010 | 0.0047* |
| | | (0.003) | (0.003) |
| Edu | | 0.0136*** | 0.0170*** |
| | | (0.004) | (0.006) |
| Market | | 0.0002 | -0.0103*** |
| | | (0.001) | (0.002) |
| Revenue | | 0.0298*** | -0.0014 |
| | | (0.009) | (0.011) |
| Intervent | | -0.2489** | -0.1487 |
| | | (0.125) | (0.136) |
| Gdp | | 0.0086 | 0.0093* |
| | | (0.007) | (0.005) |
| Fdi | | 0.0049*** | 0.0071*** |
| | | (0.002) | (0.002) |
| Road | | -0.0274*** | -0.0244*** |
| | | (0.003) | (0.003) |
| Constant | 0.0950*** | -0.2232*** | -0.4990*** |
| | (0.005) | (0.052) | (0.132) |
| Year | No | No | Yes |
| Id | No | No | Yes |
| Observations | 572 | 572 | 572 |
| R-squared | 0.725 | 0.866 | 0.947 |

Likewise, dimensional regressions are performed on the three sub-dimensions of the hinterland economic system to examine the impact of port green competitiveness on the total volume, structure, and quality of the hinterland economy. The regression results in Table 10 reveal that the regression coefficients of the explanatory variables are only significantly positive when the total volume and quality of the hinterland economy are the dependent variables. When the structure of the hinterland economy is the dependent variable, the estimated coefficients are positive but not significant. This indicates that the improvement of port green competitiveness can significantly enhance the total volume and quality of the hinterland economy, while the optimizing effect on the structure of the hinterland economy is not significant. This conclusion is consistent with the baseline regression.

### 4.4 Heterogeneity analysis

Given the numerous ports in China, each with its unique characteristics and development status, ports within the same port group often share similar resource endowments and development situations. Additionally, ports in close proximity may have overlapping hinterlands. Therefore, the entire sample is divided spatially into five major port groups as the Bohai Rim, Yangtze River Delta, Southeast Coast, Pearl River Delta, and Southwest Coast to conduct

**Table 8. Regression results of the impact of port green competitiveness on sub-dimensions of the hinterland economic system (after changing dependent variables).**

| Variable | (1) | (2) | (3) |
|---|---|---|---|
| | HlT | HlS | HlQ |
| PG | 0.6986*** | 0.1414*** | 0.3907*** |
| | (0.032) | (0.047) | (0.021) |
| Popula | 0.1644*** | -0.1334*** | 0.0783*** |
| | (0.014) | (0.016) | (0.009) |
| Science | 0.0070* | -0.0060 | 0.0061** |
| | (0.004) | (0.004) | (0.003) |
| Edu | 0.0322*** | 0.0123 | -0.0129** |
| | (0.008) | (0.009) | (0.005) |
| Market | -0.0107*** | -0.0085** | -0.0104*** |
| | (0.003) | (0.004) | (0.002) |
| Revenue | -0.0283* | 0.0774*** | 0.0105 |
| | (0.015) | (0.017) | (0.010) |
| Intervent | -0.1622 | -0.1257 | -0.1331 |
| | (0.190) | (0.216) | (0.125) |
| Gdp | 0.0097 | 0.0078 | 0.0093** |
| | (0.007) | (0.008) | (0.004) |
| Fdi | 0.0094*** | 0.0104*** | 0.0003 |
| | (0.002) | (0.003) | (0.002) |
| Road | -0.0349*** | -0.0012 | -0.0155*** |
| | (0.005) | (0.005) | (0.003) |
| Constant | -0.7648*** | -0.0249 | -0.2049* |
| | (0.184) | (0.209) | (0.121) |
| Year | Yes | Yes | Yes |
| Id | Yes | Yes | Yes |
| Observations | 572 | 572 | 572 |
| R-squared | 0.932 | 0.857 | 0.943 |

separate regressions, clarifying the regional differences in the impact of port green competitiveness on the hinterland economy. Table 11 lists the grouped regression results of port green competitiveness on the economic hinterlands of the five major port groups, revealing that the green competitiveness of ports has a significant positive effect on the economic hinterlands of the Yangtze River Delta and Pearl River Delta, but the impact on the Bohai Rim, Southeast Coast, and Southwest Coast is not significant.

## 5 Conclusion

This paper has delved into the impact of port green competitiveness on the hinterland economy, aiming to provide theoretical support and policy recommendations for the sustainable development of ports. As global trade continues to grow, ports, as key nodes in the global trade network, are increasingly playing a pivotal role in driving the hinterland economy. However, the resource consumption and environmental pollution issues associated with traditional port development models have also garnered widespread attention from the international community. Against this backdrop, this paper constructed an evaluation index system for port green competitiveness, employing methods such as the super-efficiency SBM model, entropy-

**Table 9. Regression results of the impact of port green competitiveness on the hinterland economic system (after changing core explanatory variable).**

| Variable | (1) | (2) | (3) |
|---|---|---|---|
| | Hinterland | Hinterland | Hinterland |
| PG2 | 0.0103** | 0.0100** | 0.0078* |
| | (0.005) | (0.005) | (0.004) |
| Popula | | -0.0349*** | 0.0820*** |
| | | (0.005) | (0.017) |
| Science | | -0.0094*** | 0.0073* |
| | | (0.003) | (0.004) |
| Edu | | 0.0227*** | 0.0091 |
| | | (0.006) | (0.010) |
| Market | | 0.0006 | -0.0098*** |
| | | (0.001) | (0.003) |
| Revenue | | -0.2179*** | -0.2181*** |
| | | (0.024) | (0.025) |
| Revenue | | 3.0243*** | 2.3821*** |
| | | (0.263) | (0.279) |
| Gdp | | 0.2915*** | 0.2289*** |
| | | (0.026) | (0.026) |
| Fdi | | 0.0051** | 0.0066*** |
| | | (0.002) | (0.002) |
| Road | | -0.0129*** | -0.0215*** |
| | | (0.004) | (0.004) |
| Constant | 0.0489** | 0.9438*** | 0.9628*** |
| | (0.021) | (0.151) | (0.211) |
| Year | No | No | Yes |
| Id | No | No | Yes |
| Observations | 494 | 494 | 494 |
| R-squared | 0.967 | 0.833 | 0.933 |

weighted TOPSIS, and benchmark regression models to systematically analyze the green competitiveness of ports and its interaction with the hinterland economy. By constructing an evaluation index system that includes input indicators, output indicators, and undesired output indicators, we comprehensively measured the green development status of Chinese ports and found that over the past two decades, the green competitiveness of Chinese ports has generally trended upwards, although the growth rates vary among different ports. The Yangtze River Delta port group stands out particularly, while the Bohai Rim, Southeast Coast, Pearl River Delta, and Southwest Coast port groups exhibit relatively lower green competitiveness. This suggests that the green competitiveness of ports is influenced by a variety of factors, including economic strength, geographical location, policy environment, capital investment, and technology adoption.

Meanwhile, the entropy-weighted TOPSIS method was used to quantitatively measure the hinterland economic system, providing insight into the development of the hinterland economy. The benchmark regression results indicate that the improvement of port green competitiveness has a significant positive impact on the hinterland economy. Specifically, the enhancement of port green competitiveness can significantly optimize the hinterland economic system, especially in terms of increasing the total volume and improving the quality of

**Table 10. Regression results of the impact of port green competitiveness on sub-dimensions of the hinterland economic system (after changing core explanatory variable).**

| Variable | (1) | (2) | (3) |
|---|---|---|---|
| | HlT | HlS | HlQ |
| PG2 | 0.0057* | 0.0001 | 0.0119* |
| | (0.003) | (0.010) | (0.007) |
| Popula | 0.1464*** | -0.0630*** | 0.0285* |
| | (0.024) | (0.024) | (0.015) |
| Science | 0.0100* | -0.0037 | 0.0081** |
| | (0.005) | (0.005) | (0.003) |
| Edu | 0.0235* | 0.0286** | -0.0330*** |
| | (0.014) | (0.013) | (0.008) |
| Market | -0.0101** | -0.0118*** | -0.0081*** |
| | (0.004) | (0.004) | (0.003) |
| Revenue | -0.3216*** | 0.1726*** | -0.2240*** |
| | (0.035) | (0.034) | (0.022) |
| Intervent | 3.2841*** | -1.2054*** | 2.5397*** |
| | (0.391) | (0.385) | (0.241) |
| Gdp | 0.3082*** | -0.1410*** | 0.2746*** |
| | (0.037) | (0.036) | (0.023) |
| Fdi | 0.0092*** | 0.0083*** | 0.0001 |
| | (0.003) | (0.003) | (0.002) |
| Road | -0.0313*** | 0.0002 | -0.0132*** |
| | (0.006) | (0.006) | (0.004) |
| Constant | 1.1901*** | -0.7818*** | 1.4937*** |
| | (0.295) | (0.290) | (0.182) |
| Year | Yes | Yes | Yes |
| Id | Yes | Yes | Yes |
| Observations | 494 | 494 | 494 |
| R-squared | 0.919 | 0.814 | 0.936 |

the hinterland economy. Furthermore, this paper reveals that the impact of port green competitiveness on the economic hinterlands of different regional port groups exhibits heterogeneity, with the Yangtze River Delta and Pearl River Delta port groups experiencing more significant benefits, while the impact of port green competitiveness on the Bohai Rim, Southeast Coast, and Southwest Coast is not apparent.

The research presented in this paper offers a new perspective and insights for the green development of ports. Policy makers should pay attention to the green competitiveness of ports and promote the coordinated development of ports and the hinterland economy by optimizing port management, enhancing green service capabilities, and other measures. However, we did not verify the assumptions of linear regression, such as homoscedasticity, linearity, absence of multicollinearity, and absence of outliers. Notably, our initial analysis suggests potential multicollinearity issues in the first model (refer to Table 9), which warrants further investigation in future research.Overall, this paper not only enriches the theory of the relationship between port competitiveness and the hinterland economy but also provides empirical evidence and policy recommendations for the sustainable development of ports and the hinterland economy.

**Table 11. Heterogeneity analysis of the impact of port green competitiveness on the economic hinterlands of the five major port groups.**

| Variable | (1) | (2) | (3) | (4) | (5) |
|---|---|---|---|---|---|
| | Bohai Rim | Yangtze River Delta | Southeast Coast | Pearl River Delta | Southwest Coast |
| PG | -0.0095 | 0.0086* | 0.0004 | 0.0907** | -0.0036 |
| | (0.016) | (0.005) | (0.004) | (0.036) | (0.011) |
| Popula | 0.1329 | 0.3073*** | -0.0925*** | 0.3238*** | -0.1143 |
| | (0.106) | (0.096) | (0.025) | (0.092) | (0.143) |
| Science | -0.0013 | 0.0324*** | 0.0068 | 0.0090 | -0.0085 |
| | (0.007) | (0.009) | (0.006) | (0.010) | (0.005) |
| Edu | 0.0119 | -0.0021 | 0.0014 | -0.0304 | -0.0054 |
| | (0.021) | (0.019) | (0.018) | (0.054) | (0.031) |
| Market | -0.0124* | -0.0232*** | -0.0067* | -0.0218* | -0.0045 |
| | (0.007) | (0.005) | (0.003) | (0.012) | (0.009) |
| Revenue | -0.2910*** | -0.3879*** | 0.1073*** | -0.4052*** | 0.0142 |
| | (0.049) | (0.063) | (0.031) | (0.106) | (0.073) |
| Intervent | 3.1257*** | 4.4476*** | -1.1378*** | 5.7643*** | -0.0999 |
| | (0.575) | (0.653) | (0.211) | (1.172) | (0.926) |
| Gdp | 0.4478*** | 0.2389*** | -0.0486* | 0.4894*** | 0.0049 |
| | (0.047) | (0.074) | (0.024) | (0.105) | (0.085) |
| Fdi | 0.0028 | -0.0012 | 0.0334*** | 0.0171** | -0.0035 |
| | (0.006) | (0.006) | (0.007) | (0.008) | (0.003) |
| Road | -0.0068 | -0.0230* | 0.0058 | -0.0165 | 0.0033 |
| | (0.011) | (0.012) | (0.006) | (0.025) | (0.007) |
| Constant | -0.2552 | 1.8490** | -0.8895** | 1.5046** | 0.6669 |
| | (0.855) | (0.787) | (0.341) | (0.597) | (0.853) |
| Year | Yes | Yes | Yes | Yes | Yes |
| Id | Yes | Yes | Yes | Yes | Yes |
| Observations | 152 | 171 | 38 | 76 | 57 |
| R-squared | 0.925 | 0.968 | 0.999 | 0.980 | 0.931 |

## Acknowledgments

We would like to express our sincere gratitude to the reviewer for their valuable insights and constructive feedback, which have significantly contributed to the improvement of our manuscript.

## Author Contributions

**Conceptualization:** Yajing Li, Ming Yin.

**Data curation:** Yajing Li.

**Formal analysis:** Yajing Li.

**Funding acquisition:** Ming Yin.

**Methodology:** Yajing Li.

**Project administration:** Ming Yin.

**Supervision:** Ming Yin.

**Writing – original draft:** Ming Yin, Jiawei Ge.

**Writing – review & editing:** Jiawei Ge.

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
