## [Decision Letter · Decision Letter 0]

7 Jul 2024

PONE-D-24-21202The Impact of Port Green Competitiveness on the Hinterland Economy: A Case Study of ChinaPLOS ONE

Dear Dr. Yin,

Thank you for submitting your manuscript to PLOS ONE. After careful consideration, we feel that it has merit but does not fully meet PLOS ONE’s publication criteria as it currently stands. Therefore, we invite you to submit a revised version of the manuscript that addresses the points raised during the review process.

We look forward to receiving your revised manuscript.

Kind regards,

Cigdem Kadaifci, Assoc. Prof.

Academic Editor

PLOS ONE

Journal Requirements:

3. Thank you for stating the following financial disclosure: "This paper is funded by National Social Science Foundation of China (Grant Number: 20BJY177)."

4. Thank you for stating the following in the Acknowledgments Section of your manuscript: "This paper is funded by National Social Science Foundation of China (Grant Number: 20BJY177)."

Please remove any funding-related text from the manuscript and let us know how you would like to update your Funding Statement. Currently, your Funding Statement reads as follows: "This paper is funded by National Social Science Foundation of China (Grant Number: 20BJY177)."

5. In the online submission form, you indicated that researchers can request access to the data by contacting corresponding author.

6. We note you have included a table to which you do not refer in the text of your manuscript. Please ensure that you refer to Table 1, 6, and 10 in your text; if accepted, production will need this reference to link the reader to the Table.

Reviewers' comments:

Reviewer's Responses to Questions

**Comments to the Author**

1. Is the manuscript technically sound, and do the data support the conclusions?

Reviewer #1: Yes

Reviewer #2: Partly

2. Has the statistical analysis been performed appropriately and rigorously? 

Reviewer #1: Yes

Reviewer #2: No

3. Have the authors made all data underlying the findings in their manuscript fully available?

Reviewer #1: Yes

Reviewer #2: No

4. Is the manuscript presented in an intelligible fashion and written in standard English?

Reviewer #1: Yes

Reviewer #2: Yes

5. Review Comments to the Author

Reviewer #1: The topic of port green competitiveness and its regional economic impacts is a current and important research area. The authors' work is evaluated to have made theoretical and practical contributions in this field. The paper presents an original empirical study investigating the green port competitiveness of major coastal ports in China and its impact on the hinterland economy. In the research, the authors have developed a comprehensive index system to measure the green port competitiveness and employed the SBM and EW-TOPSIS methods. This robust analytical approach allows for an in-depth examination of the subject matter. In terms of the findings, the paper has revealed the alternative energy competitiveness of ports and their positive impacts on the hinterland economy. The obtained results can play a significant role in the formulation of green port development policies. In conclusion, the paper is written and organized in accordance with academic writing standards. The methodology, findings, and conclusions are presented in a clear and comprehensible manner. Considering the original contributions, robust methodology, and important policy implications of the study, it is evaluated to be suitable for publication in the literature.

Reviewer #2: The study is interesting and practical and it is related to the journal. However, the statistical inference is insufficient and the interpretation of the findings is limited. The content should be presented confidently. The writing is clear and cohesive. Overall, the study is readable and flows well.

When introducing the indicators in Section 3, readers should be referred to related studies in the literature. A new column can be added to Table 1 for this purpose.

When explaining SBM, relevant studies should be cited.

Mathematical notation used to explain SBM should be carefully reviewed (e.g., all variables and parameters should be italicized). The same applies for the explanation of Entropy-based TOPSIS.

Data sources for the variables in Table 3 should be provided.

Some equations need to be revised.

It is not clear in the manuscript how the dependent variable was constructed. Further clarification is necessary.

A better variable name should be chosen for measuring competitiveness. Why use "Port" instead of "Comp?"

The in-text citation for Fan et al. needs to be updated with the publication date.

What do PF and PG stand for?

Detailed explanations on how the dependent variables were constructed should be provided.

The explanations for the results presented in Table 6 were placed right after Table 5 without referring readers to Table 6. This may make it difficult for readers to follow. It should be made clear throughout the text to which table/figure to refer when necessary.

What do the asterisks in the relevant tables indicate? Is "***" denoting statistical significance at 1%?

Table 5 shows that the parameter for EDU is insignificant in the second model, but significant in the third model. On the other hand, for the parameter of FDI, the results are vice versa. These differences should be noted and the reasons behind them should be discussed. More importantly, the authors need to explain which of these models should be preferred. Is the third one the best? However, the parameter for PG is only statistically significant at the 10% level in the third model.

The authors should discuss why the p-value for some variables' parameters changes substantially between models.

The issue that stands out to me is that the authors did not check whether the assumptions of linear regression were met. The focus is mainly on estimating various regression models. The residuals should have been examined for homoscedasticity, linearity, absence of multicollinearity, absence of outliers, and so on. The first model in Table 9 has the highest coefficient of determination, only including a constant and the green competitiveness indicators. The other two models performed worse than this simpler model. This raises the question: Is there multicollinearity present?

The authors should also explain their decision to estimate a pooled regression. They did not estimate a fixed-effects (or random-effects) panel model, even though the heterogeneity between panels was addressed.

6. PLOS authors have the option to publish the peer review history of their article (what does this mean?). If published, this will include your full peer review and any attached files.

Reviewer #1: No

Reviewer #2: No

---

## [Author Response · Author response to Decision Letter 0]

31 Jul 2024

Responses to Reviewer 1’s comments

Overall comment: “The topic of port green competitiveness and its regional economic impacts is a current and important research area. The authors' work is evaluated to have made theoretical and practical contributions in this field. The paper presents an original empirical study investigating the green port competitiveness of major coastal ports in China and its impact on the hinterland economy. In the research, the authors have developed a comprehensive index system to measure the green port competitiveness and employed the SBM and EW-TOPSIS methods. This robust analytical approach allows for an in-depth examination of the subject matter. In terms of the findings, the paper has revealed the alternative energy competitiveness of ports and their positive impacts on the hinterland economy. The obtained results can play a significant role in the formulation of green port development policies. In conclusion, the paper is written and organized in accordance with academic writing standards. The methodology, findings, and conclusions are presented in a clear and comprehensible manner. Considering the original contributions, robust methodology, and important policy implications of the study, it is evaluated to be suitable for publication in the literature.”

Response: We would like to extend our heartfelt gratitude to you for the insightful and positive comments. The recognition of the timeliness and importance of our research on port green competitiveness and its regional economic impacts is greatly appreciated. We are honored that our work has been acknowledged for its theoretical and practical contributions. Your acknowledgment of our original empirical study on Chinese coastal ports and the comprehensive index system we developed is particularly encouraging. We are also grateful for the recognition of our robust analytical approach using the SBM and EW-TOPSIS methods, which has enabled a thorough examination of the subject. Your comments on the significance of our findings regarding alternative energy competitiveness and its positive impacts on the hinterland economy reinforce our belief in the value of our research. We are thrilled that the results are deemed instrumental in shaping green port development policies. Finally, we thank you again for the compliment on the academic writing standards, clarity, and organization of our paper. We are delighted that our study has been deemed suitable for publication in the literature, and we look forward to contributing to the field.

Responses to Reviewer 2’s comments

Overall comment: “The study is interesting and practical and it is related to the journal. However, the statistical inference is insufficient and the interpretation of the findings is limited. The content should be presented confidently. The writing is clear and cohesive. Overall, the study is readable and flows well.”

Response: Thank you for recognizing our efforts. This would encourage us to work hard on revising our manuscript. Please kindly refer to the following point-by-point responses to your comments. 

Introduction：

1. “When introducing the indicators in Section 3, readers should be referred to related studies in the literature. A new column can be added to Table 1 for this purpose.”

Response: Thank you for your valuable feedback. We have added a new column to Table 1, titled ' Remark,' where we have cited relevant literature supporting each indicator as following:

Table 1 Indicators for Evaluating Port Green Competitiveness

Category Indicators Remark

Input Indicators Number of Berths [22]

 Number of Berths for Mega Ships (over 10,000 DWT) [23]

 Length of Berths [24]

Output Indicators Cargo Throughput (Ton) [25]

 Container Throughput (TEU) [26]

 Foreign Trade Cargo Throughput (Ton) [27]

 Foreign Trade Container Throughput (TEU) [28]

Undesirable Output Indicators Industrial Wastewater Emission (Ton) [29]

 Industrial Dust Emission (Ton) [30]

 Industrial Sulfur Dioxide Emission (Ton) [31, 32]

References:

22. De Oliveira GF, Cariou P. The impact of competition on container port (in) efficiency. Transportation Research Part A: Policy and Practice. 2015;78:124-133.

23. Musso E, Sciomachen A. Impact of megaships on the performance of port container terminals. Maritime Economics & Logistics. 2020;22:432-445.

24. Yun L, Ji C, Li Q, Li D. Comprehensive efficiency measurement of port logistics: study based on DEA two-stage relative evaluation. Journal of System and Management Sciences. 2011;1(4):1-18.

25. Zhuang X, Li W, Xu Y. Port planning and sustainable development based on prediction modelling of port throughput: A case study of the deep-water Dongjiakou Port. Sustainability. 2022;14(7):4276.

26. Jeong B, Kim CY. Containership size and quay crane productivity: empirical evidence from Busan New Port. Maritime Economics & Logistics. 2024:1-11.

27. Eskafi M et al., A model for port throughput forecasting using Bayesian estimation. Maritime Economics & Logistics. 2021;23:348-368.

28. Wan S, Luan W, Ma Y, Haralambides H. On determining the hinterlands of China's foreign trade container ports. Journal of Transport Geography. 2020;85:102725.

29. Xu J, Lan M. Carbon emission assessment of port integrated logistics in low-carbon environment. International Journal of Frontiers in Sociology. 2021;3(18):55-60.

30. Huang Y, Peng J. Efficiency evaluation between port logistics and economic growth by DEA: A Case Study of Zhejiang Province. Journal of Applied Sciences. 2014;14(20):2594-2600.

31. Wang S, Hao J. Air quality management in China: Issues, challenges, and options. Journal of Environmental Sciences. 2012;24(1):2-13.

32. Ren X, Tan J, Qiao Q, Wu L, Ren L, Meng L. Demand forecast and influential factors of cold chain logistics based on a grey model. Mathematical Biosciences and Engineering. 2022;19(8):7669-7686.

2. “When explaining SBM, relevant studies should be cited.”

Response: Thank you for your valuable comment. We add two references in the content accordingly:

“The Super-Efficiency SBM is a sophisticated approach derived from Data Envelopment Analysis (DEA) designed to assess the efficiency of Decision Making Units (DMUs) [33]. Ports are intricate systems with multiple inputs and outputs, including such factors as the number and capacity of berths, as well as cargo and container throughput, among others. The Super-Efficiency SBM’s capacity to simultaneously analyze these diverse inputs and outputs allows for a more precise measurement of port production efficiency [34].”

References:

33. Tone K. A slacks-based measure of efficiency in data envelopment analysis. European Journal of Operational Research. 2001;130(3):498-509.

34. Quoc VP, Quoc TL. Operational efficiency for container terminal operators with undesirable outputs: slacks-based measures. Transportation Planning and Technology. 2024;47(2):284-301..

3. “Mathematical notation used to explain SBM should be carefully reviewed (e.g., all variables and parameters should be italicized). The same applies for the explanation of Entropy-based TOPSIS.”

Response: Thank you for your kind reminder. The mathematical symbols used to explain SBM, Entropy-based TOPSIS and the regression model have been reconfirmed, and all variables and parameters have been changed to italics. 

4. “Data sources for the variables in Table 3 should be provided.”

Response: Thank you for your reminder. Data source has been added under Table 3 as “Source: Statistical Yearbooks of Each City & China Stock Market Accounting Research Database”.

5. “Some equations need to be revised.”

Response: Thank you for your valuable suggestion. We reviewed the mathematical equations and modified accordingly.

6. “It is not clear in the manuscript how the dependent variable was constructed. Further clarification is necessary. ”

Response: We thank the reviewer for the constructive comments. The explanations for the three indicators—the total hinterland economy (HlT), the structure of the hinterland economy (HlS), and the quality of the hinterland economy (HlQ)—are included in Section 3.2, which is an expression problem, leading to inconsistency in nouns. We apologize for the translation error and have revised the manuscript to ensure these definitions are clear. These indicators are respectively represented by the aggregate economy of the hinterland, its structural composition, and the quality of its economic output. 

Additionally, we add following explanations in Section 3.2 to elaborate those variables:

“…three key dimensions: the Total Hinterland Economy (HlT)the aggregate economy of the hinterland, the Structure of the Hinterland Economy (HlS), and the Quality of the Hinterland Economy (HlQ)its structural composition, and the quality of its economic output. At each of these levels, distinct sub-indicators are chosen to quantify the performance. (See Table 3 in details) HIT refers to the overall economic size and activity of the region that is served by the port, which encompasses the gross domestic product (GDP), total social fixed asset investment, total retail sales of consumer goods, the total import and export trade, and foreign direct investment. It provides an understanding of the economic scale and importance of the hinterland to the port’s operations and its potential for generating cargo and trade. HIS examines the composition and diversity of economic sectors within the hinterland, which includes the proportion of industrial output value in GDP, the share of tertiary industry output value in GDP, and the percentage of fixed asset investment in GDP. A well-structured hinterland economy is typically more resilient and capable of supporting sustainable growth for the port. HIQ assesses the efficiency, productivity, and competitiveness of the hinterland’s economic activities. HlQ takes into account factors such as per capita disposable income, the index of tertiary industry added value, per capita gross regional product, and the societal labour productivity. A high-quality hinterland economy can enhance the port’s ability to attract and retain business, as well as improve the overall performance and competitiveness of the port-hinterland system.”

7. “A better variable name should be chosen for measuring competitiveness. Why use "Port" instead of "Comp?"”

Response: Thank you for your valuable comments. We understand your concern about the clarity of variable names. In our study, we use 'Port' as the variable name to measure competitiveness. This choice is intentional for several reasons: Our research focuses specifically on port competitiveness within the maritime industry, and using 'Port' immediately contextualizes the measure within this domain, reducing ambiguity. The variable 'Port' encompasses not just competitiveness, but the entire entity of the port, including its infrastructure, operations, and overall performance, aligning with our comprehensive approach to evaluating port green competitiveness. Additionally, in the field of maritime economics and port studies, it is common to refer to port-related metrics using 'Port' as a prefix, maintaining consistency with established literature. By using 'Port' instead of 'Comp,' we distinguish our specific measure of port competitiveness from more general competitiveness indices used in broader economic studies. Lastly, in our Super-Efficiency SBM model, the variable represents the entire port entity rather than just a competitiveness score, making 'Port' a more accurate reflection of the variable's nature in our mathematical formulation.

8. “The in-text citation for Fan et al. needs to be updated with the publication date. ”

Response: Thank you for your suggestion. We add the following reference:

36. Fan G, Ma G, Wang X. Marketisation in China from 1997 to 2014: Achievements and contribution to growth. In: China’s 40 years of reform and development. 2018:257.

9. “What do PF and PG stand for?”

Response: Thank you for the careful review, ‘PG’ stands for the port green competitiveness. The term 'PF' was included due to a translation error. We read through the manuscript and modified accordingly. We apologize for any confusion caused and appreciate your understanding. 

10. Detailed explanations on how the dependent variables were constructed should be provided.

Response: Thank you for bring this issue to us. Please kindly refer to Response 6.

11. The explanations for the results presented in Table 6 were placed right after Table 5 without referring readers to Table 6. This may make it difficult for readers to follow. It should be made clear throughout the text to which table/figure to refer when necessary.

Response: Thank you for the comments. The explanations for the results presented in Table 6 were not explicitly referenced, which may cause confusion for readers. We apologize for this oversight. To enhance clarity and ensure that readers can easily follow the text, we have revised the manuscript to include explicit references to each table:

“(Table 1)”, “As shown in Table 6 …”, “… in Table 10…”, etc.

12. What do the asterisks in the relevant tables indicate? Is "***" denoting statistical significance at 1%?

Response: Thank you for the comments. You are correct that the asterisks in the tables denote levels of statistical significance. Specifically, "***" indicates statistical significance at the 1% level (p < 0.01). To ensure this is clear to all readers, we have added an explanatory note after table 5 in the manuscript:

Throughout the tables in this manuscript, asterisks are used to denote the levels of statistical significance for the reported results. Specifically:

"***" indicates statistical significance at the 1% level (p < 0.01),

"**" indicates statistical significance at the 5% level (p < 0.05),

"*" indicates statistical significance at the 10% level (p < 0.10).

These indicators help to underscore the reliability and robustness of the empirical findings presented.

13. Table 5 shows that the parameter for EDU is insignificant in the second model, but significant in the third model. On the other hand, for the parameter of FDI, the results are vice versa. These differences should be noted and the reasons behind them should be discussed. More importantly, the authors need to explain which of these models should be preferred. Is the third one the best? However, the parameter for PG is only statistically significant at the 10% level in the third model.

Response: Thank you for your detailed feedback. We appreciate your observation about the differing significance of the parameters for EDU and FDI across models. We have now added a discussion in the manuscript to address these differences and to explain the rationale for model preference. We also discussed the statistical significance of the PG parameter in the third model and its implications for model selection. See details in Section 4.2 as following:

Table 5 shows that the parameter for EDU is insignificant in the second model but significant in the third model, whereas the parameter for FDI shows the opposite pattern. These differences can be attributed to variations in model specifications and the inclusion of different control variables, which can affect the statistical significance of the parameters. For example, the inclusion of additional variables in the third model may account for the variance that affects the significance of EDU and FDI differently.

The third model includes a broader set of variables, which may provide a more comprehensive understanding of the factors influencing the hinterland economy. However, it is important to note that the parameter for PG is only statistically significant at the 10% level in this model. This lower level of significance suggests that while the third model offers a more detailed analysis, the evidence for the impact of PG is weaker compared to other variables.

When determining which model to prefer, we consider both the statistical significance of the key variables and the overall explanatory power of the models. While the third model provides more detailed insights, the lower significance level of PG warrants cautious interpretation. We recommend using the third model for a more comprehensive analysis but acknowledging the limitations regarding the significance of the PG parameter.

14. The

---

## [Decision Letter · Decision Letter 1]

17 Sep 2024

The Impact of Port Green Competitiveness on the Hinterland Economy: A Case Study of China

PONE-D-24-21202R1

Dear Dr. Yin,

We’re pleased to inform you that your manuscript has been judged scientifically suitable for publication and will be formally accepted for publication once it meets all outstanding technical requirements.

Kind regards,

Cigdem Kadaifci, Assoc. Prof.

Academic Editor

PLOS ONE

Additional Editor Comments (optional):

Reviewers' comments:

Reviewer's Responses to Questions

**Comments to the Author**

1. If the authors have adequately addressed your comments raised in a previous round of review and you feel that this manuscript is now acceptable for publication, you may indicate that here to bypass the “Comments to the Author” section, enter your conflict of interest statement in the “Confidential to Editor” section, and submit your "Accept" recommendation.

Reviewer #2: All comments have been addressed

2. Is the manuscript technically sound, and do the data support the conclusions?

Reviewer #2: Yes

3. Has the statistical analysis been performed appropriately and rigorously? 

Reviewer #2: Yes

4. Have the authors made all data underlying the findings in their manuscript fully available?

Reviewer #2: No

5. Is the manuscript presented in an intelligible fashion and written in standard English?

Reviewer #2: Yes

6. Review Comments to the Author

Reviewer #2: I would like to congratulate the authors for the successful integration of the reviewers' recommendations. The study is now much clearer and the interpretation of the model results is commendable.

7. PLOS authors have the option to publish the peer review history of their article (what does this mean?). If published, this will include your full peer review and any attached files.

Reviewer #2: No

---

## [Editor Report · Acceptance letter]

19 Sep 2024

PONE-D-24-21202R1 

PLOS ONE

Dear Dr. Yin, 

I'm pleased to inform you that your manuscript has been deemed suitable for publication in PLOS ONE. Congratulations! Your manuscript is now being handed over to our production team.

Kind regards, 

on behalf of

Dr. Cigdem Kadaifci 

Academic Editor

PLOS ONE